# Evolutionary transition from a single RNA replicator to a multiple replicator network

Ryo Mizuuchi [1,2✉], Taro Furubayashi[3] & Norikazu Ichihashi [1,4,5✉]

In prebiotic evolution, self-replicating molecules are believed to have evolved into complex living systems by expanding their information and functions open-endedly. Theoretically, such evolutionary complexification could occur through successive appearance of novel replicators that interact with one another to form replication networks. Here we perform long-term evolution experiments of RNA that replicates using a self-encoded RNA replicase. The RNA diversifies into multiple coexisting host and parasite lineages, whose frequencies in the population initially fluctuate and gradually stabilize. The final population, comprising five RNA lineages, forms a replicator network with diverse interactions, including cooperation to help the replication of all other members. These results support the capability of molecular replicators to spontaneously develop complexity through Darwinian evolution, a critical step for the emergence of life.

[1] Komaba Institute for Science, The University of Tokyo, Meguro, Tokyo 153-8902, Japan. [2] JST, PRESTO, Kawaguchi, Saitama 332-0012, Japan. [3] Department of Applied Chemistry, Graduate School of Engineering, The University of Tokyo, Bunkyo, Tokyo 113-8656, Japan. [4] Department of Life Science, Graduate School of Arts and Science, The University of Tokyo, Meguro, Tokyo 153-8902, Japan. [5] Universal Biology Institute, The University of Tokyo, Meguro, Tokyo 153-8902, Japan. ✉email: mizuuchi@bio.c.u-tokyo.ac.jp; ichihashi@bio.c.u-tokyo.ac.jp

An origins-of-life scenario depicts Darwinian evolution from self-replicating molecules, such as RNA, toward complex living systems[1–3]. How molecular replicators could develop complexity by continuously expanding information and functions is a central issue in prebiotic evolution[4,5]. An expected route for complexification is that novel RNA replicators successively emerged and co-replicated so that increased genetic information can be stored at a population level, before their assembly into a long genome[4–8]. Although several theoretical studies investigated the possibility of complexification[9–11] and stable coexistence[12–16] of molecular replicators, empirical demonstration has been challenging.

To date, diverse molecular replicators have been constructed based on biomaterials such as DNA, RNA, and peptides[17–21]. Although considerable efforts have been made to design interactions among these replicators[22–26], spontaneous complexification was generally precluded due to their inability to undergo Darwinian evolution through continuous mutation accumulation and natural selection. The study by Ellinger et al.[23] was an exception, but they used a predefined replicator network to initiate evolution, which was also limited to the short-term. Thus, it has remained an open question whether a single molecular replicator can evolve into a complex replicator network.

Previously, we constructed an RNA that replicates using an RNA replicase translated from itself[27]. During replication, mutations are introduced, and occasional recombination deletes replicase-encoded regions to generate a parasitic RNA that replicates by exploiting replicases derived from other RNAs (replicase-encoding "host" RNAs). The RNAs can undergo Darwinian evolution in a serial transfer experiment, and a previous attempt (120 rounds, 600 h) demonstrated the successive appearance of new host and parasitic RNA lineages showing defense and counter-defense properties[28,29]. However, these lineages dominated the population in turn and only transiently, possibly due to a short evolutionary timescale.

Here, we continued the serial transfer experiment up to 240 rounds (1200 h). Sequence analysis uncovered that two previously detected host RNA lineages became sustained and further diverged into multiple sublineages of host and parasitic RNAs. The population dynamics of each lineage gradually changed during the evolution, from dynamically fluctuating stages to quasi-stable coexistence, suggesting the appearance of co-replicative relationships among the lineages. Biochemical analyses supported the co-replication of dominant RNAs in the different lineages containing a cooperative RNA that replicates all other members, thus establishing a multiple replicator network.

## Results

**Long-term evolution of an RNA replicator**. The RNA replication system (Fig. 1a) consists of a single-stranded RNA (host RNA) that encodes the catalytic subunit of Qβ replicase (an RNA-dependent RNA polymerase) and a reconstituted *Escherichia coli* translation system[30]. RNA replication occurs through the translation of the replicase subunit, which becomes active in association with elongation factors Tu and Ts (EF-Tu and EF-Ts) in the translation system. The introduction of mutations into the host RNA during replication generates host RNA variants with different features. Spontaneous RNA recombination deletes a part of the replicase gene to generate parasitic RNAs while retaining 5′ and 3′ terminal sequences for recognition by the replicase. When encapsulated in micro-sized water-in-oil droplets, host RNAs can sustainably replicate in the presence of parasitic RNAs[28].

We performed a long-term replication experiment using the RNA replication system, started with a clonal host RNA population (a round 128 clone of the previous study[27]). In the

experiment, we repetitively (1) performed RNA replication by incubating the replication system in water-in-oil droplets at 37 °C for 5 h, (2) diluted the droplets 5-fold with new droplets containing the fresh translation system, and (3) induced the fusion and division of the droplets by vigorous stirring to mix the contents well (Fig. 1b). After every replication step, we measured average host RNA concentrations by quantitative PCR after reverse transcription (RT-qPCR) using primers that specifically bind to host RNAs (Fig. 1c). The parasitic RNAs could not be uniquely targeted by RT-qPCR because of their varied deletion sites; therefore, we measured their concentrations from the band intensities after polyacrylamide gel electrophoresis (Supplementary Fig. S1). We conducted 120 rounds of the transfer cycle (rounds 1–120) in the previous studies[28,29] and additional 120 rounds (rounds 121–240) in this study.

The population dynamics of the host and parasitic RNAs showed multiple differences between the early (rounds 1–120) and late (rounds 121–240) stages (Fig. 1c). First, the irregularly changing host and parasitic RNA concentrations in the early stage turned into a relatively regular oscillation in the late stage. Second, parasitic RNAs with different lengths (~ 220 nt, ~1100 nt, and ~500 nt) newly appeared only in the early stage, as reported previously[29], and ~500 nt parasitic RNAs dominated the population throughout the late stage, although a small amount of ~220 nt RNAs was detected transiently (at rounds 136–140, 147–151, 178–181, and 207–210). These results suggested that the mode of evolution changed in the late stage.

**Sequence analysis**. To understand the evolutionary dynamics, we analyzed host and parasitic RNA sequences throughout the long-term replication experiment. For rounds 1–115, we used the data obtained previously[29]. For rounds 116–240, we recovered RNA mixtures at 18 points and subjected host and ~500 nt parasitic RNAs to PacBio sequencing. For parasitic RNAs, we analyzed only ~500 nt ones, first detected at round 115, because the other sizes of parasitic RNAs were rarely detected in the late stage. From 680 to 10000 reads of host and parasitic RNAs at rounds 116–240 (Supplementary Table S1), we identified 111 dominant mutations that were detected in at least 10% of the reads at any round, consisting of 74 and 30 unique mutations for host and parasitic RNAs, respectively, and 7 common mutations. We here focused only on the dominant mutations and ignored minor mutations introduced during error-prone replication of Qβ replicase and PacBio sequencing. Next, we defined consensus genotypes as the combinations of the 111 dominant mutations and used these genotypes in the following analysis.

To examine the evolutionary trajectory, we performed phylogenetic analysis using the three most frequent host and parasitic RNA genotypes in each sequenced round. The phylogenetic tree is displayed together with the round-by-round frequency and dominant mutations of each genotype (Fig. 2). The host and parasitic RNA lineages are represented by thick and thin lines, respectively, and the genotypes detected in the final population at round 237 were indicated with black stars. The ancestral host RNA (Ancestor) initially accumulated mutations (ancestral lineage, HL0) and then diverged into two host RNA lineages (HL1 and HL2), corresponding to those containing Host-99 and Host-115 RNAs detected in the previous study[29], respectively. In HL1, host RNAs further accumulated mutations, and top genotypes were successively replaced until the final rounds. In HL2, fewer mutations were accumulated than in HL1, corresponding to short horizontal branches in the tree, and some genotypes (e.g., HL2–228 and HL2-155) remained dominant in the last 100 rounds. Other RNAs in HL2 accumulated independent mutations around round 129 and formed another

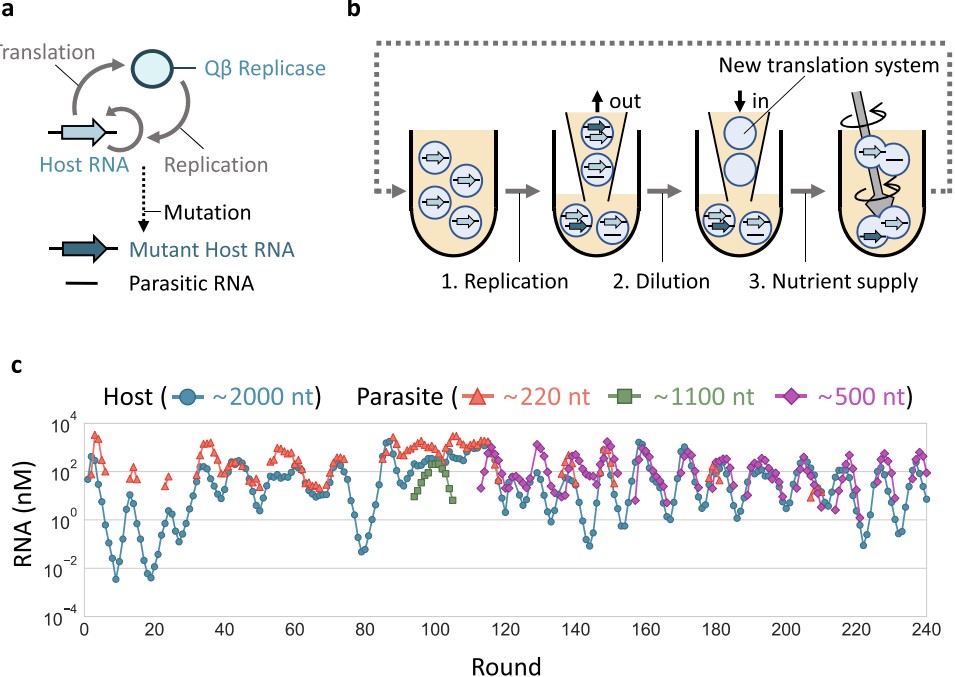

**Fig. 1 Long-term replication experiment. a** The RNA replication system. The original host RNA replicates via translation of the self-encoded replicase, by which mutant host RNAs and parasitic RNAs could be generated. **b** Schematic representation of long-term replication experiments in water-in-oil droplets. (1) RNA replication was performed at 37 °C for 5 h. (2) Droplets were 5-fold diluted with new droplets containing the translation system. (3) Droplets were vigorously mixed to induce their random fusion and division. **c** Concentration changes of host and parasitic RNAs of different lengths. Host RNA concentrations were measured by RT-qPCR, and parasitic RNA concentrations were measured from corresponding band intensities after gel electrophoresis. Parasitic RNA concentrations were not plotted in rounds where they were undetectable.

set of branches, named HL3, which contains a genotype (HL3–228) that remained in the final round with at least 10 distinct mutations and also exhibited markedly different characteristics from those in HL2 as shown later.

For parasitic RNAs, two lineages (PL1 and PL2) originated from HL2. PL1, corresponding to parasite-γ in the previous study[29], diverged from HL2 around round 115 but was not detected in the last 80 rounds. PL2 emerged around the same round and accumulated unique mutations, and several genotypes (e.g., PL2-228) remained in the final round. From PL2, a sublineage of parasitic RNAs appeared, named PL3, which contains a genotype (PL3–228) that accumulated no less than four novel mutations and displayed unique replication abilities as shown later.

**Population dynamics of the lineages**. Next, we analyzed the population dynamics of each host and parasitic RNA lineage using the 100 most frequent genotypes of all host and parasitic RNAs at each sequenced round. These genotypes covered 46–100% and 80–100% of the total host and parasitic RNA reads of each round, respectively. The genotypes in different lineages show distinct patterns and rates of mutation accumulation (Supplementary Figs. S3 and S4), supporting the unique evolutionary history of each lineage.

We then plotted the frequencies of each lineage for every sequenced round (Fig. 3). For the host RNA lineages (Fig. 3b), the ancestral lineage, HL0, gradually became less frequent from round 72. Instead, the frequencies of HL1, HL2, and HL3 increased but highly fluctuated ranging from the undetected level to almost 100% until round 190. Afterward, the frequencies of all three evolved lineages became relatively stable as they were persistently detected at more than 3% of the population. For the parasitic RNA lineages (Fig. 3c), PL1 was initially dominant but

quickly became rare. In contrast, PL2 dominated the population throughout the rounds with more than 10% of the population. PL3 became dominant at round 190 and then coexisted with PL2 at similar frequencies. Overall, the five RNA lineages, HL1, HL2, HL3, PL2, and PL3, were initially rare or highly fluctuated in frequency but shifted to relatively stable coexistence in the last ~50 rounds of the experiment, as illustrated in Fig. 3a.

To investigate the reproducibility of the diversification and coexistence of several host and parasite lineages observed above, we performed two additional long-term replication experiments (Supplementary Text 1 and Figs. S5–10). Initiated with the droplet mixture at round 76 of the main experiment (Fig. 1c), we performed independent 164 rounds (total 240 rounds) of serial transfer in each experiment and analyzed RNA sequences using the same method. As in the main experiment, we observed similar gradual diversification and relatively stable coexistence of multiple host and parasitic RNA lineages, despite the accumulation of different mutations in each additional experiment.

**Development of a multiple replicator network**. The transition to the coexistence of diversified RNA lineages raised the possibility that RNAs in each lineage replicated interdependently. To examine this possibility, we collected the ancestral clone at round 0 (HL0–0) and the most dominant RNA clones in the five sustained lineages (HL1, HL2, HL3, PL2, PL3) at rounds 120, 155–158, and 228: 3 clones at round 120 (HL1-, HL2-, and PL2–120), 4 clones at round 155–158 (HL1–158, HL2-, HL3-, and PL2–155), and 5 clones at round 228 (HL1-, HL2-, HL3-, PL2-, and PL3–228), all of which accumulated different sets of mutations (Supplementary Fig. S11). The RNA clones were termed as the name of corresponding lineages followed by rounds at which the clone was most dominant in the lineages.

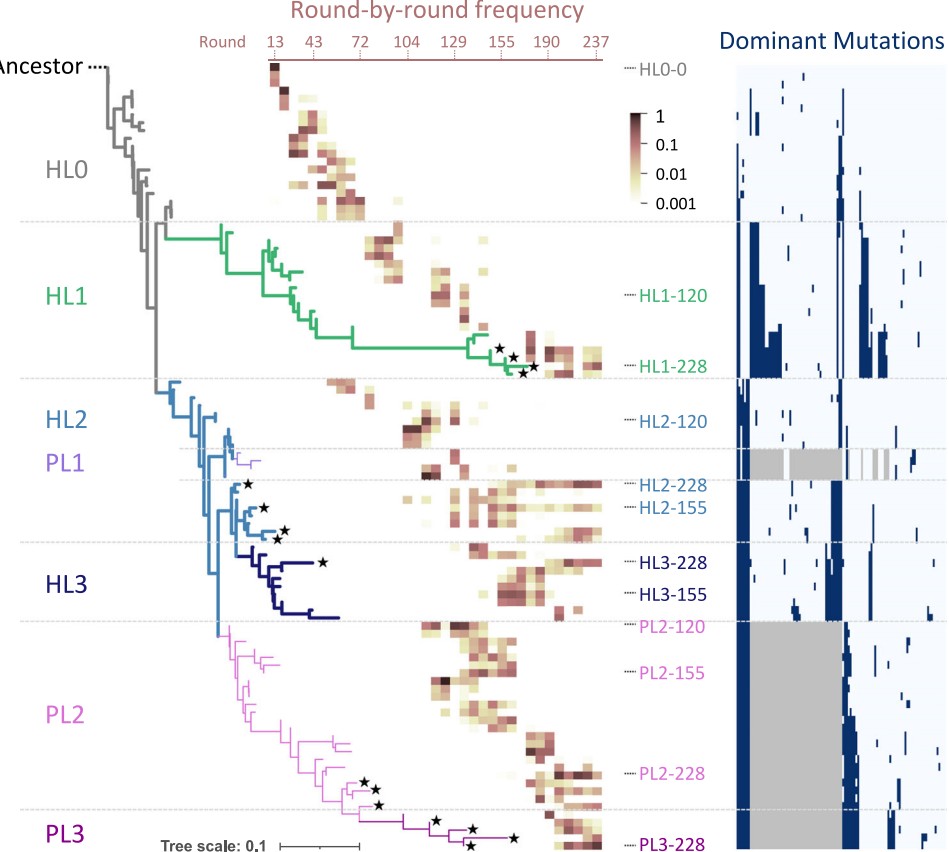

**Fig. 2 Phylogeny of consensus host and parasitic RNA genotypes.** The three most frequent host and parasitic RNA genotypes in all sequenced rounds are shown, with the ancestral host RNA ("Ancestor") designated as the root of the tree. Branches comprising the defined lineages are colored differently. Host (HL0–3) and parasitic (PL1–3) RNA lineages are shown as thick and thin lines, respectively. The heatmap superimposed on the tree shows the frequencies of each genotype in total host or parasitic RNA reads over all sequenced rounds (from left to right). Black star shapes at the tips of branches mark genotypes that remained in the last sequenced round. Genotypes used for biochemical analysis are indicated with the names of the corresponding RNA clones if presented in the tree. The list of dominant mutations is shown on the right; navy and gray colors indicate the presence of a point mutation and deletion, respectively. An enlarged view of the list is presented in Supplementary Fig. S2.

To determine replication relationships among these RNA clones, that is, the extent of each RNA replication by each host-encoded replicase, we performed translation-uncoupled replication experiments in the following two steps (Fig. 4a). First, we incubated one of the host RNA clones (RNA 1) at 37 °C for 2 h to induce the translation of the replicase but in the absence of UTP to preclude RNA replication. Next, we initiated replication by incubating the reaction mixture that contains RNA 1 and the synthesized replicase in the presence of UTP at 37 °C for 2 h, with or without the same concentration of another host or parasitic RNA clone (RNA 2) of the selected rounds. We then measured the extent of replication for each RNA clone by RT-qPCR with sequence-specific primers (Fig. 4b–e). We visualized the interdependent replication abilities of the selected RNAs as directed graphs (Fig. 3d). An RNA clone is represented by a node (names were abbreviated to lineage names), and RNA replication is indicated by an arrow pointing from a host RNA that produced the replicase to a replicated RNA. The arrow widths correspond to the extent of replication (see Supplementary Fig. S12 for details of graph representation).

The graphs indicate the transition of RNA replication relationships through the long-term replication experiment. At round 0, the ancestral RNA (HL0) was replicated by the self-encoded replicase ("HL0 replicase"). At round 120, two host (HL1 and HL2) and one parasitic (PL2) RNAs appeared, but their interaction was limited. Both HL1 and HL2 replicated with their

respective replicases without detectable interdependency. HL2 replicase also replicated PL2 as efficiently as HL2, whereas HL1 replicase did not. At rounds 155–158, another host RNA lineage (HL3) appeared, and the replication relationship became complicated. HL1 and HL2 were mainly replicated by their respective replicases, whereas HL3 was primarily replicated by HL2 replicase. PL2 was also preferentially replicated by HL2 replicase. At round 228, another parasitic RNA lineage (PL3) appeared, and the replication relationship became even more complicated. HL1 acquired the ability to utilize HL2 replicase, as well as the self-encoded replicase. HL2 replicase could replicate not only HL2 but also all other RNAs. HL3 replication still largely relied on HL2, although the dependence was weakened compared to rounds 155–158. In parasitic RNA clones, PL2 was only replicated by HL2 replicase, and its replication by the other replicases became negligible. In contrast, PL3 was replicated by all three replicases similarly, but none of the replications was as efficient as that of PL2 by HL2 replicase, indicating that PL2 is an HL2-specific parasite, whereas PL3 is a general parasite. These results demonstrated that the RNAs in each lineage have different biochemical properties, and the replication relationship among the RNAs gradually changed as evolution proceeded to form a larger replicator network.

To characterize the biochemical properties underlying the observed replication relationship, we further examined the five RNA clones at round 228 (HL1-, HL2-, HL3-, PL2-, and

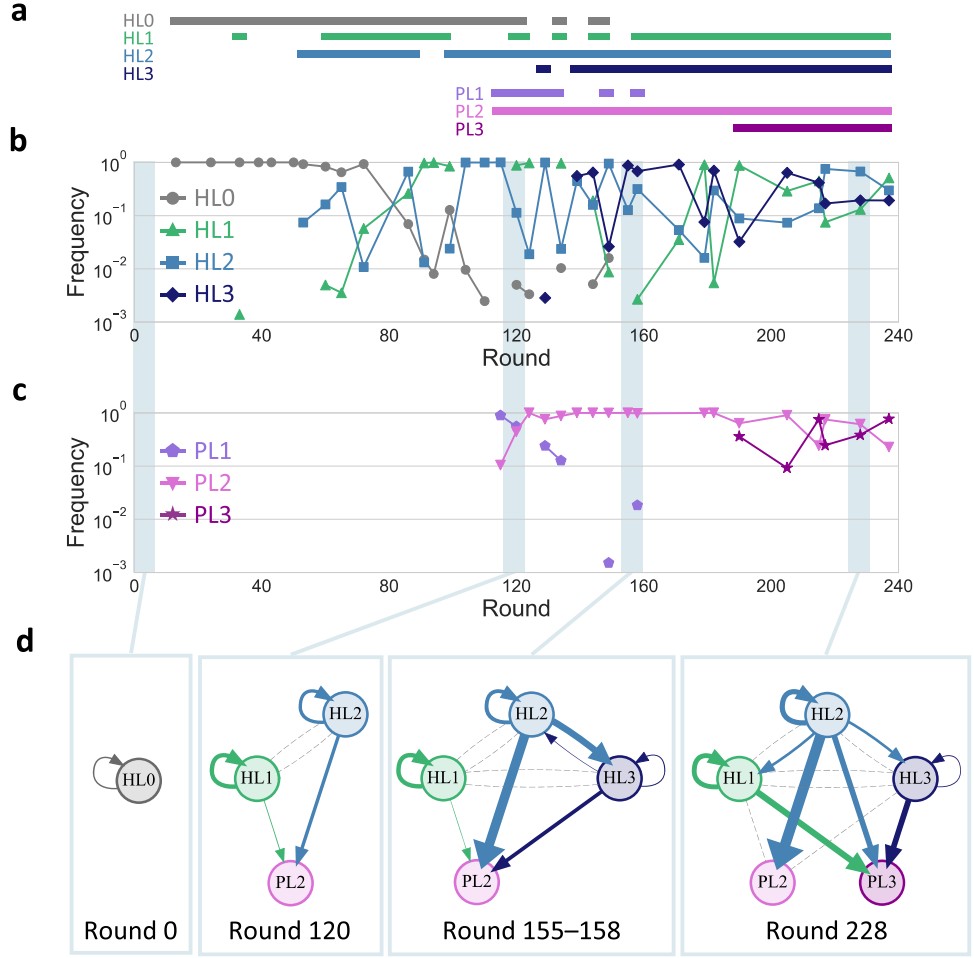

**Fig. 3 Development of replicator networks. a–c** Frequencies of the lineages in total sequence reads of the analyzed genotypes for the host (**b**) and parasitic (**c**) RNAs, with horizontal lines above the graphs (**a**) indicating rounds where the frequency of each lineage is more than 0.1 %. **d** Directed graphs based on translation-uncoupled experiments (Fig. 4) for representing inter-dependent replication of the RNA clones of each selected round. Nodes represent RNA clones of indicated rounds. Arrows indicate direct replication with widths proportional to binary logarithm of the measured levels of relative replication. Gray dashed lines indicate undetected replication events from host RNA clones.

PL3–228) by fully decoupling translation and replication reactions. We first analyzed the synthesis of each encoded replicase during translation and found that HL1- and HL3–228 showed approximately twice as much replicase synthesis than HL2–228, whereas PL2- and PL3–228 did not show detectable translation activity (Supplementary Fig. S13). Next, we replicated the RNA clones using purified replicases derived from HL1-, HL2-, and HL3–228 (Supplementary Fig. S14) and found that the tendency of replication was mostly consistent with that of the translation-uncoupled replication experiment (Fig. 4e). For example, HL1- and HL3–228 replicases preferentially replicated their corresponding host RNAs (HL1- and HL3–238, respectively) and PL3–228 among parasitic RNAs, whereas HL2–228 replicase replicated all five RNA clones. These results indicate that the interdependent RNA replication at round 228 (Fig. 3d) can be mainly explained by the different template specificities of the three evolved replicases. The change in template specificity possibly relied on the different properties of replicases and RNAs, such as secondary structures (Supplementary Figs. S15 and S16).

It should be noted that although we focused on interactions between two RNAs, there may be higher-order interactions that arise only when an RNA is replicated by multiple RNAs simultaneously. However, we confirmed that such higher-order interactions played only a minor role in the replicator network (Supplementary Text 2, Figs. S17 and S18).

We also note that the experiments described above measured total RNA replication without separating the plus and minus strands. The efficiency of replication can vary depending on which strand is used as a template. Therefore, we measured the synthesis of plus and minus strands separately for the five RNA clones at round 228 and found that plus strands were more synthesized than minus strands for any of the RNAs (Supplementary Fig. S19).

**Sustained co-replication of multiple RNAs**. To examine whether the selected RNA clones could reproduce the co-replication dynamics of each lineage in the late stage of the long-term evolution experiment (Fig. 3b. c), we initiated a serial transfer experiment with 10 nM each of the five RNA clones at round 228 (HL1-, HL2-, HL3-, PL2-, and PL3–228). HL3–228 was soon diluted out, whereas the other four RNAs sustainably replicated for at least 22 rounds (Fig. 5a). A different initial condition (10 nM host RNAs and 0.1 nM parasitic RNAs) consistently showed sustained replication of the same four RNAs for the same rounds (Supplementary Fig. S20a).

Next, to understand whether the replication relationships between these RNA clones (Fig. 3d) could sufficiently explain the observed replication dynamics, we created a theoretical model of the five RNAs whose replication rates were determined from average replication levels of HL1-, HL2-, HL3-, PL2-, and

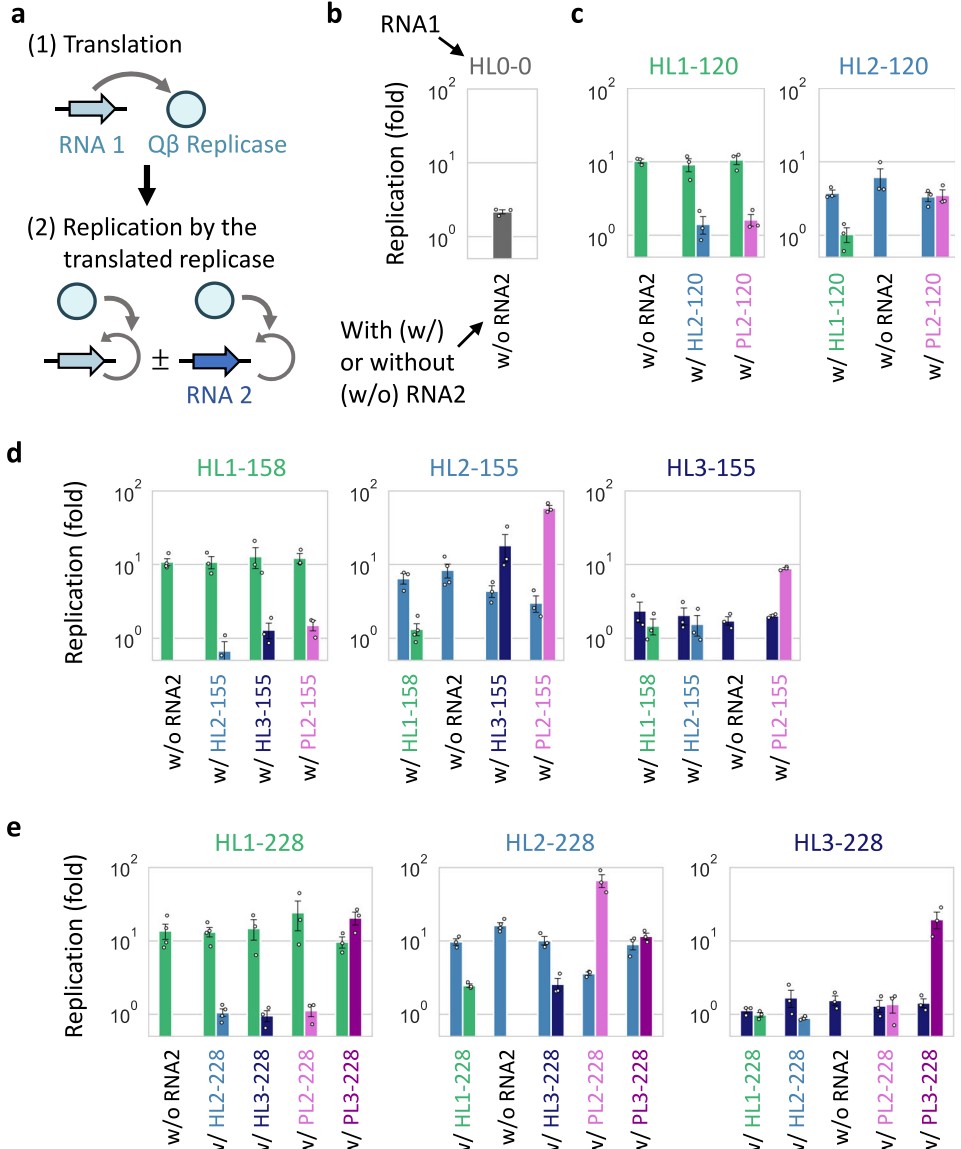

**Fig. 4 Translation-uncoupled replication experiments. a** The experiments were performed in two steps. (1) First, one of the host RNA clones (RNA 1, 30 nM) was incubated at 37 °C for 2 h to translate the replicase without RNA replication. (2) The reaction mixture was diluted, and RNA replication was initiated with the translated replicase at 37 °C for 2 h while stopping translation, in the presence or absence of the same concentration (10 nM) of another RNA clone (RNA 2). The replication of each RNA was measured by sequence-specific RT-qPCR. **b–e** Replication of one or pairs of RNA clones at rounds 0 (**b**), 120 (**c**), 155–158 (**d**), and 228 (**e**). Colors match those in Fig. 3d. Error bars indicate mean ± SEM ($n = 3$ or 4 as shown as individual data points). Measurements were taken from distinct samples. Average fold replications (>1.5-fold) were used to draw the directed graphs (Fig. 3d). Source data are provided as a Source data file.

PL3–228 in the translation-uncoupled RNA replication experiments (Fig. 4). We simulated continuous replication in uniform-sized compartments by modeling the three steps shown in Fig. 1b. Consistent with the experimental results, four RNAs, except for one based on HL3-228, sustainably co-replicated by displaying similar concentration dynamics (Fig. 5b). These experimental and simulation results demonstrated that the replication relationship among the selected RNA clones explain the sustainable replication of at least four RNA lineages.

Finally, we examined how the four RNAs could sustainably replicate when competing with each other. Hypothesizing that each RNA helped sustain the replication of others, we simulated continuous replication by removing each of the four RNAs from the five-member network. We found that the absence of HL1-, HL2-, PL2-, and PL3–228, tended to cause the extinction of PL3-,

PL2-, HL1-, and HL2- and PL2-228, respectively (Supplementary Fig. S21). A serial transfer experiment in the absence of HL2–228 reproduced the simulated replication dynamics in that PL2-228 was diluted out (Supplementary Fig. S20b). These results support the interdependence of the four RNAs, facilitating the coexistence of multiple replicators. Of note, the experiments in the absence of the other RNAs (HL1-, PL2-, and PL3–228) were unsuccessful because the removed RNAs appeared soon, probably due to the presence of contamination at an undetected level and/or de novo production through recombination and mutations.

## Discussion

Our results demonstrate an evolutionary transition scenario of molecular replicators from a single common ancestor to a multi-

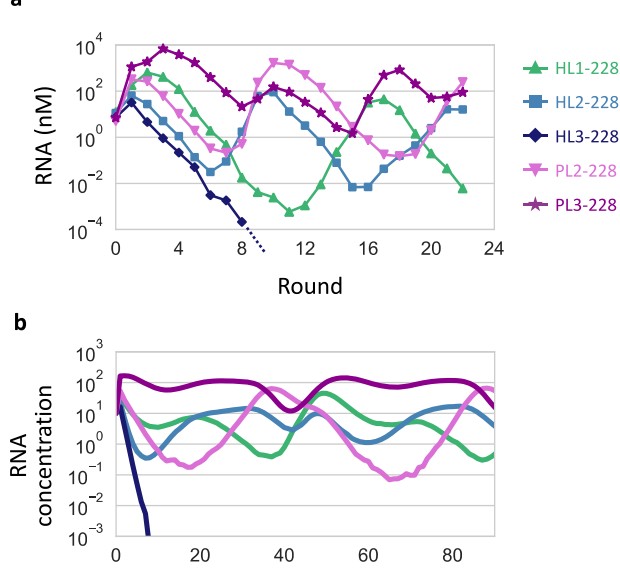

**Fig. 5 Co-replication dynamics of the evolved RNA clones. a** RNA concentration changes in a long-term replication experiment started with 10 nM each of the five RNA clones at round 228, measured by sequence-specific RT-qPCR. **b** Representative RNA concentration changes in a simulated long-term replication using the theoretical model. The replication rates of the simulated RNAs were based on those of the RNA clones of the same colors in (**a**).

membered network. During long-term replication, a clonal host RNA population gradually diversified into multiple host and parasitic RNA lineages. Their frequencies initially fluctuated probably due to competition, but stabilized as the evolution progressed (Fig. 3a–c). The co-replication of isolated RNAs in five lineages exhibited the gradual development of replicator networks, eventually consisting of five types of RNAs with distinct features (Fig. 3d). We confirmed that four of the RNAs sustainably co-replicated both in the experiment and simulation (Fig. 5). This observation exemplifies the possible appearance and coexistence of a diverse set of replicators despite common resources (e.g., NTP), a primary concern in origins-of-life research[8]. Our results provide evidence that Darwinian evolution drives complexification of molecular replicators, paving the way toward the emergence of living systems.

One intriguing point in the developed replicator networks is the cooperativity of HL2-155 and −228, which replicate three of the four and all five dominant replicators in the associated networks, respectively (Fig. 3d). Such cooperation could unite distinct replicators and facilitate the emergence of complex biological systems[7]. The round-by-round frequencies in Fig. 2 confirmed the maintenance of the cooperative RNAs (HL2-155 and -228) for more than 80 rounds, indicating the stability of cooperative replication strategies. Although previous studies demonstrated the maintenance of cooperative traits in the presence of compartments[27,31], the spontaneous advent of cooperators has been elusive. Our study suggests that cooperative replicators could emerge and easily become dominant in a realistic molecular replication system.

The mechanism that sustained the replication of multiple RNAs is not obvious from the structure of the replicator network (Fig. 3d) because it lacks direct reciprocal interactions between replicators, which have been employed in both simulations and experiments to demonstrate the coexistence of multiple molecular replicators[12–16,31]. One key factor may be a compartment that

could prevent the complete domination of parasites in the population[28,32,33]. Moreover, our simulation displayed that removing one RNA replicator typically drove other RNAs to extinction (Supplementary Fig. S21), partly supported by experiments (Supplementary Fig. S20b), indicating that the RNAs in the network may have sustained each other indirectly. The mechanisms behind these observations could be explained as follows. (1) Removal of HL1−228 resulted in the disappearance of PL3−228 through competition with PL2−228, which adapted to HL2−228. (2) Removal of HL2−228 caused the extinction of PL2−228 because it could not replicate in the absence of HL2−228. (3) Removal of PL2-228 led to the competitive exclusion of HL1−228 by HL2−228, which replicated slightly faster and was more resistant to PL3−228. (4) Removal of PL3−228 made HL1−228 outcompete HL2−228 because the remaining parasitic RNA (PL2-228) parasitized only HL2−228 among the host RNAs. The disappearance of HL2−228 then caused the extinction of PL2-228, as its replication relied on HL2−228. Overall, all RNAs aided the replication balance of each other, and thus, the long-term coexistence.

A remaining mystery is the unsustainable HL3−228 replication in the transfer experiment with the isolated RNAs (Fig. 5) despite its sustainability in the main evolution experiment (Fig. 2). This discrepancy may be associated with a missing RNA that was not isolated as a dominant RNA but helped HL3−228 replication in the evolution experiment. Thus, further complex replicator networks may have formed during the evolution experiment.

We note that in the RNA replication system, the translation process could affect RNA replication. For example, the competition between the ribosome and Qβ replicase to use plus RNA strands in opposite directions may inhibit the synthesis of minus strands by replication, as previously suggested for a host RNA[34]. Such competition may have biased the replication of RNA clones at round 228 to the plus-strand synthesis (Supplementary Fig. S19).

In the long-term replication, several RNA lineages appeared through coevolution between host and parasitic RNAs. Parasitic replicators are believed to have constantly appeared and acted as drivers for the evolution of complex systems at many levels of biological organization, ever since before life originated[35–37]. Notably, computational models demonstrated that parasitic replicators that specifically parasitize a certain host replicator generated a new ecological niche where new host replicators that defended against the parasitism evolved and coexisted[11,38]. In our experiment, the maintenance of multiple host RNA lineages may have been associated with a similar interplay between host and parasitic RNAs rather than the presence of diverse resources or spatial niches, other factors that could cause such adaptive radiations[39,40]. For example, HL1 and HL2 at round 228 are relatively resistant to different parasites, PL2 and PL3, respectively (Fig. 3d). Our study indicates that the diversification and complexification through coevolution with parasitic replicators is a plausible scenario for a realistic replication system consisting of RNA and proteins.

In the present study, we defined parasitic RNAs as shortened sequences that deleted the replicase gene and can replicate only in the presence of a replicase-encoding RNA (host RNA). Similarly, some of the host RNAs in the population may produce dysfunctional replicases and replicate only by utilizing active replicases translated from other host RNAs. Our previous transfer experiment showed that approximately 60% of a host RNA population did not produce functional replicases due to random mutations[31]. Although we focused only on dominant mutations in this study, heterogeneity in replicase activity may be an important factor to consider for a comprehensive understanding of the evolutionary dynamics.

Long-term experimental evolution is a powerful methodology that has provided us with fundamental insights into the principle of evolution[41–44]. Laboratory evolution of bacterial, eukaryotic, and viral systems has also demonstrated host-parasite coevolution and diversification[39,45–51], similar to those observed in this study. The simplicity of our RNA replication system, compared with biological organisms, allows us to examine evolutionary events with unprecedented resolution. For example, a small set of defined components enables the detailed characterization of replication strategies for each species (e.g., Fig. 4, Supplementary Figs. S13 and S14). We can also readily obtain a large number ($>10^5$) of the entire genome sequences of all replicating species (Supplementary Table S1) for extensive investigation of population genetics throughout evolution, which is challenging for living organisms because of the much larger genomes[42,43]. Thus, our simple experimental setup offers a unique approach to deeply look into evolutionary phenomena.

## Methods

**Plasmids and RNAs.** Two plasmids, each encoding the ancestral host RNA (HL0–0) and HL2–120, were obtained previously as the plasmids encoding round 128 clone[27] and Host-115[29], respectively. Two plasmids, each encoding HL3–155 and PL2–120, were constructed using gene synthesis service of Eurofins Genomics. Nine plasmids, each encoding one of the other evolved host and parasitic RNA clones, were constructed by site-specific mutagenesis of plasmids obtained in this study or previously[29] using primers that contained each RNA clone-specific mutation, as described in Supplementary Table S2. All RNA clones were prepared from the plasmids by in vitro transcription with T7 RNA polymerase (Takara) after digestion with Sma I (Takara). The remaining plasmids were treated with DNase I (Takara), and the transcribed RNAs were purified using the RNeasy Mini kit (Qiagen). The RNA sequences are available in Supplementary Data 1.

**Long-term replication experiment.** In the main long-term replication experiment (Fig. 1c), the first 120 rounds of replication were performed in previous studies, started with a clonal population of HL0–0[28,29], and additional 120 rounds (total 240 rounds) of replication were performed by the same method in this study, started with the RNA population in round 120. In the two additional long-term replication experiments (E2 and E3, Supplementary Fig. S5), 164 rounds of replication were performed, started with the RNA population in round 74 of the main experiment (total 240 rounds). In round 1, 10 µl of reaction mixture containing 1 nM HL0–0 and the translation system was vigorously mixed with 1 ml of buffer-saturated oil using a homogenizer (POLYTRON PT-1300D, KINEMA-TICA) at 16,000 rpm for 1 min on ice to prepare water-in-oil droplets. The preparation of the translation system (based on the reconstituted *Escherichia coli* translation system[30]) and buffer-saturated oil was described previously[27]. Next, the droplets were incubated at 37 °C for 5 h to induce RNA replication through protein translation. From round 2 to 240, 200 µl of water-in-oil droplets in the previous round, 10 µl of the translation system, and 800 µl of buffer-saturated oil were homogenized by the same method to prepare a new droplet population, followed by incubation under the same condition (at 37 °C for 5 h) to induce RNA replication. The concentrations of host and parasitic RNAs were determined after replication. Host RNA concentrations were measured by RT-qPCR with Mx3005P Real-Time PCR System (Agilent technologies) or QuantStudio 3 Real-Time PCR System (Thermo Fisher Scientific) after diluting the droplets 100-fold with 1 mM EDTA (pH 8.0), by using One Step TB Green PrimeScript PLUS RT-PCR Kit (Takara) and host RNA-specific primers (Supplementary Table S3). Dilution series of the original host RNA was used to draw a standard curve. For the measurement of parasitic RNA concentrations, the droplets were recovered by centrifugation (22,000 × g, 5 min). The recovered solution was mixed with four volumes of diethyl ether, centrifuged (11,000 × g, 1 min) to remove the diethyl ether phase, and purified using RNeasy Mini kit (Qiagen). Obtained RNA samples were then subjected to 8% polyacrylamide gel electrophoresis in 1× TBE buffer. Fluorescence intensities of parasitic RNA bands were quantified after staining with SYBR Green II (Takara) and visualized using FUSION-SL4 (Vilber-Lourmat), and concentrations were determined based on a dilution series of standard RNA bands. For long-term replication of isolated RNA clones (Fig. 5a, Supplementary Fig. S20), the mixture of 10 nM or 0.1 nM of the four or five clones at round 228 was used as the initial RNAs, and the experiments were performed as described above. Dilution series of each RNA was used to draw a standard curve for the measurement of RNA concentrations.

**Sequence analysis.** The RNA samples of rounds 120, 124, 129, 134, 139, 144, 149, 155, 158, 171, 179, 182, 190, 205, 215, 217, 228, and 237 (the main experiment); rounds 92, 114, 129, 144, 160, 173, 183, 189, 195, 200, 220, and 239 (additional experiment E2); and rounds 86, 94, 105, 135, 155, 175, 198, 219, and 237

(additional experiment E3) in the long-term replication experiments were obtained as described above. It should be noted that RNA replication in round 120 of the main experiment was performed in the previous study[29], whereas the RNAs were sequenced in this study. The RNAs were reverse-transcribed with PrimeScript reverse transcriptase (Takara), and PCR-amplified with KOD FX DNA polymerase (Toyobo). Approximately 2000 bp (host) and 500 bp (parasite) PCR products were size-selected through 0.8% agarose gel-electrophoresis with E-Gel CloneWell Gels (Thermo Fisher Scientific), followed by purification with MinElute PCR Purification Kit (Qiagen). The cDNA samples were barcoded through PCR amplification with KOD FX DNA polymerase and sequenced using PacBio RS II (Pacific Biosciences, rounds 120–155 samples of the main experiment) as described previously[29] or using PacBio Sequel (Pacific Biosciences, the other samples) at Macrogen Japan. The obtained circular consensus sequences (CCS) reads were quality-filtered as described previously (PacBio RS II)[29], or CCS reads with more than Q20 were retained (PacBio Sequel). Host or parasite CCS reads at each round (Supplementary Table S1) were aligned with HL0–0 as the reference sequence using MAFFT v7.450 with the FFT-NS-2 algorithm[52], and mutations that were present in more than 10% of the reads at each round of host or parasite sequences were identified as dominant mutations.

**Genotype analysis.** Consensus genotypes of host and parasitic RNAs were constructed as the combination of the dominant mutations (111, 105, and 97 mutations for the main experiment, E2, and E3, respectively) by removing uncommon mutations in each sequence (Supplementary Data 1). Genotypes were also constructed based on the same mutations from host and parasite (~500 nt) CCS reads that were previously obtained at rounds 13, 24, 33, 39, 43, 50, 53, 60, 65, 72, 86, 91, 94, 99, 104, 110, and 115 (for analysis of E2 and E3, only reads of rounds 13–72 were used). The three most dominant genotypes of host and parasitic RNAs at each round were then subjected to phylogenetic analysis. The phylogenetic trees were constructed using the neighbor-joining method via MEGA X[53] with the default parameters, and host and parasitic RNA lineages were defined based on the trees as described in the main text. The trees were visualized using Interactive Tree Of Life (iTOL)[54]. Next, to classify the 100 most dominant genotypes of host and parasitic RNAs at each round into the lineages, Hamming distances between all pairs of host RNA genotypes and those of parasitic RNA genotypes were calculated for each experiment. Based on the distance matrices, the genotypes were mapped in two-dimensional spaces (Supplementary Fig. S22) by determining the position of each genotype through Principal Coordinate Analysis for dimensional reduction, as described previously[29]. The genotypes with smaller distances in the maps were categorized in the same lineages. It should be noted that for the main experiment, there were only 65 and 88 host RNA genotypes at rounds 24 and 33, respectively, and 70, 64, and 58 parasitic RNA genotypes at rounds 115, 120, and 124, respectively.

**Translation-uncoupled replication experiments.** The experiments were performed in the following two-step reactions. First, one of the RNA clones (30 nM, RNA 1) was incubated at 37 °C for 2 h in the translation system, without UTP to preclude RNA replication. Second, an aliquot of the first reaction was 3-fold diluted in the fresh translation system containing 1.25 mM UTP and further incubated at 37 °C for 2 h with or without another RNA clone (10 nM, RNA 2), in the presence of 30 µg/ml streptomycin to inhibit further translation. After 0 and 2 h incubation of the second reaction, the concentrations of RNA clones were measured by RT-qPCR using sequence-specific primers (Supplementary Table S3) after 10,000-fold dilution with 1 mM EDTA (pH 8.0) as described above, and fold replications were determined. To determine the fold replication of RNA2, measured fold replications were divided by those of negative control reactions performed with only RNA2. The average of these replications was used to depict directed graphs (Fig. 3d) as represented in Supplementary Fig. S12.

**Analysis of protein translation by sodium dodecyl sulfate polyacrylamide gel electrophoresis (SDS-PAGE).** An RNA clone (300 nM) was incubated at 37 °C for 2 h in the translation system and FluoroTect GreenLys tRNA (Promega), without UTP to preclude RNA replication. After translation, an aliquot was treated with 0.1 mg/ml RNase A (Qiagen) at 37 °C for 15 min, incubated at 95 °C for 4 min in SDS sample buffer [50 mM tris(hydroxymethyl)aminomethane hydrochloride (Tris-HCl, pH 7.4), 2% SDS, 0.86 M 2-mercaptoethanol, and 10% glycerol], and subjected to 10% SDS-PAGE. The synthesized fluorescently labeled proteins were visualized using FUSION-SL4 (Vilber-Lourmat).

**RNA replication by purified Qβ replicase.** Qβ replicase of each RNA clone (HL1-228, HL2-228, and HL3-228) was purified as described in the previous study[55]. Briefly, an encoded replicase subunit was co-expressed with EF-Tu and EF-Ts in *Escherichia coli*, and then, the cell lysate was subjected to ammonium sulfate precipitation, followed by anion and cation exchange chromatography. Purified replicases were analyzed by 10% SDS-PAGE and Coomassie Brilliant Blue staining. For the replication reaction, 10 nM of an RNA clone was replicated at 37 °C for 2 h using 10 nM of each purified replicase. The reaction was performed under the same conditions for the replication reaction in translation-uncoupled

replication experiments, and the fold replication of each RNA clone was determined accordingly.

**Translation-coupled replication experiments**. One, two, or three RNA clones of choice (10 nM each) were incubated at 37 °C for 5 h in the translation system. After 0, 2, and 5 h incubation, the concentrations of RNA clones were measured by RT-qPCR using sequence-specific primers (Supplementary Table S3) as described above and fold replications were determined. In some cases, the synthesized amount of plus and minus strand RNAs were separately measured by quantitative PCR using TB Green Premix Ex Taq II (Tli RNaseH Plus) (Takara) after reverse transcription with PrimeScript reverse transcriptase (Takara) and strand-specific primers (Supplementary Table S3).

**Bahadur expansion analysis**. The mathematical detail of the analysis was described in the previous study[56]. For each combination of three RNA clones (RNA$_i$, RNA$_j$, and RNA$_k$) examined in the translation-coupled replication experiments (Supplementary Fig. S17), we used the fold replications of each RNA (after 2 h incubation) to obtain the following equation:

$$\log_{10}\left(\text{fold replication of RNA}_i\right) = f_0 + w_j z_j + w_k z_k + w_{jk} z_j z_k \tag{1}$$

and

$$z_{j(k)} = \begin{cases} 1, \text{ if RNA}_{j(k)} \text{ is present} \\ -1, \text{ if RNA}_{j(k)} \text{ is absent} \end{cases}, \tag{2}$$

where $f_0$, $w_j$, $w_k$, $w_{jk}$ are the zeroth, first, and second order Bahadur coefficients, respectively. $f_0$ is the average fold replication of RNA$_i$, $w_j$ and $w_k$ represent the 1-body contribution of RNA$_j$ or RNA$_k$ to RNA$_i$ replication, respectively, and $w_{jk}$ represents the 2-body contribution of a set of RNA$_j$ and RNA$_k$ to RNA$_i$ replication. There are four fold replications attributed to RNA$_i$ in one combination, measured at 2 h in replication without the other RNAs, co-replication with RNA$_j$ or RNA$_k$, and co-replication with both RNA$_j$ and RNA$_k$, allowing the calculation of the four Bahadur coefficients. The coefficient of determination ($R^2$) was then calculated for each first order Bahadur coefficient ($w_x$) as follows:

$$R^2 = \frac{(w_x)^2}{\sum \left(\log_{10}\left(\text{fold replication of RNA}_i\right) - f_0\right)^2 / 4}. \tag{3}$$

If the sum of $R^2$ for $w_j$ and $w_k$ is 1, by definition, there is no interaction between RNA$_j$ and RNA$_k$ to affect RNA$_i$ replication.

**Simulation**. The previous theoretical model[31] was modified to simulate the replication dynamics of five RNAs (RNA$_i$, $1 \leq i \leq 5$) in a serial transfer format. The developed model includes the following parameters.

[RNA$_i$]: Concentration of RNA$_i$ in each compartment

$k_{ij}$: Rate constant of RNA$_i$ replication catalyzed by RNA$_j$ in each compartment ($1 \leq j \leq 5$) (Supplementary Fig. S23)

$C$: Carrying capacity (300)

$D$: Dilution rate (5)

$M$: Number of compartments (300000)

$F$: Fusion-division number (1.3)

$k_{ij}$ values for all combinations of $i$ and $j$ were determined from the average fold replications of RNA clones at round 228 obtained in the translation-uncoupled RNA replication experiments (Fig. 4e). By assuming that translation-uncoupled RNA replication is a first-order reaction, we calculated $k_{ij}$ as natural logarithm of $R_{eff}$ values, which are described in Supplementary Fig. S12. $k_{ij}$ reflects multiple activities, including the translation activity of RNA$_j$, the catalytic activity of the expressed replicase, and the ability of RNA$_i$ to act as a template for the replicase. Then we consider the simplest scheme of RNA replication, where the replication rates of each RNA depend on the carrying capacity and all types of RNAs in the same compartments as follows:

$$\frac{d[RNA_i]}{dt} = [RNA_i] \sum_{j=1}^{5} k_{ij}[RNA_j] \left(1 - \frac{\sum_{i=1}^{5}[RNA_i]}{C}\right). \tag{4}$$

A long-term replication was simulated by the following processes:

(1) 10 $M$ molecules each of four or five types of RNAs were randomly distributed in $M$ compartments according to the Poisson distribution. The sizes of compartments were assumed to be uniform and the volume was set to 1 so that the concentrations of RNAs equaled their numbers in a compartment.

(2) RNAs were replicated according to the above differential equation, solved by the Python package scipy.integrate.odeint for a fixed reaction time.

(3) The compartments were diluted $D$-fold with vacant compartments.

(4) Total RNAs in two randomly chosen compartments were mixed and redistributed into the two compartments according to the binomial distribution. This process was repeated $M \times F$ times.

(5) The processes (2) to (4) were repeated for indicated rounds.

We note that the extension of the model to explicitly incorporate replicases (Supplementary Text 4 and Fig. S24) generated similar results.

**Reporting summary**. Further information on research design is available in the Nature Research Reporting Summary linked to this article.

## Data availability

The data supporting the findings of this study are available from the corresponding authors upon reasonable request. Supplementary Data 1 contains the RNA sequences of the isolated clones and information on all analyzed genotypes. PacBio sequence data before alignment are available on Dryad[57]. Source data are provided with this paper.

## Code availability

The codes for the simulation are available at GitHub (https://git.io/JcliB).

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

## Acknowledgements

We thank Koutou Yukawa for helpful comments and Hideaki E. Kato for technical assistance. This research was supported by JSPS KAKENHI (19K23763 and 21H05867 to R.M., 15KT0080, 18H04820, and 20H04859 to N.I.), JST PRESTO (JPMJPR19KA to R.M.), Astrobiology Center Program of National Institutes of Natural Sciences (AB021005 to N.I.), and "Innovation inspired by Nature" Research Support Program, SEKISUI CHEMICAL CO., LTD. (N.I.).

## Author contributions

R.M. and N.I. designed the project. R.M. and T.F. performed the three long-term replication experiments and collected sequence data. R.M. performed sequence analysis, the other biochemical experiments, and simulation. R.M. and N.I. analyzed data and wrote the paper with comments from T.F.

## Competing interests

The authors declare no competing interests.
