## [Peer Review File · Nature Communications]

Title: Evolutionary transition from a single RNA replicator to a multiple replicator networkREVIEWER COMMENTS

Reviewer #1 (Remarks to the Author):

This manuscript has high potential. Very important is that it is an experimental approach to an important problem that was largely explored theoretically before.

There are a few burning issues that MUST be addressed:

1. How do you handle plus and minus strands? This is important because this could shed light on the evolution of strand asymmetry also for those species that are being translated (grossly in line with previous theoretical works). Can it be detected that translated strands are transcribed more efficiently from the untranslated ones by the replicase than the other way round?
2. Since we are lacking a generalized replicase ribozyme you must go through a protein replicase. This is a bit unfortunate, but currently there is no better solution. BUT an extra problem is how the RNA strands interact with the in vitro translation machinery, about which we do not know much (this is also related to the first question above). Some discussion would be in order.
3. RNA structures are badly missing, though knowledge about them might aid understanding considerably. This is true for the primary sequence, but with help of the Vienna package a tentative exploration of secondary structures is also feasible.
4. Parasites are not only variants that are translationally inert, but also those that produce dysfunctional proteins -- what about the latter in the experiment?
5. Interpretation of the emerging replicator networks and their stability is not clear enough. Since the protein enzyme is in the loop, I would have modelled the systems with RNA AND encoded replicase explicitly. (This raises questions of how the protein enzymes evolve in terms of structure--not addresses at all). This would immediately stand out as being faster transcribed, but also by a higher rate of attachment to the replicase also. This is pertinent to the "Four-minus-one" experiments also. It is obvious why removal of HL2 causes PL2 to go extinct. It is much less so why removal of parasite PL3 causes the extinction of HL2. The explanation can be that when you remove PL3 then HL1 gets a big advantage because PL3 is its sole parasite, whereas the generalist HL2 still suffers from the strong parasite PL2. The balance is tilted by the fact that in effect HL1 outcompetes HL2. Discuss!
6. Coexistence of HL1 and HL2. Could it be that self-inhibition by association of the plus and negative strands at high concentration contributes to this phenomenon? Also, if you attempt to take replicases explicitly into account, you are advised to look at

I.R. Epstein (1979) Competitive coexistence of self-reproducing macromolecules. *J Theor Biol* 78(2):271-98. doi: 10.1016/0022-5193(79)90269-8.

I suggest major revision.

Reviewer #2 (Remarks to the Author):

Mizuuchi et al. have continued their previous study (*Nat. Commun.* 4:2494, 2013) of the co-evolution of

Q β RNA and corresponding RNA-encoded Q β replicase protein, carried out within droplets of a water-in-oil emulsion. Adding to the previous 120 rounds of growth and dilution, they now have completed a total of 240 rounds, and have observed increasingly complex population dynamics among the set of replicating species. Over the course of evolution, there are a succession of “host” replicators that retain the activity of the replicase protein, as well as “parasitic” replicators that are truncated RNAs that do not encode a functional protein but are replicated by protein that is produced by a host. This work is reminiscent of that of Lenski and colleagues concerning the longitudinal evolution of *E. coli*, but Mizuuchi et al. employ a purely biochemical system that affords more precise understanding of the properties of the competing species over time. Most notably, their study demonstrates the spontaneous emergence of cooperative, as well as competitive, replicators. The scientific quality of the work is outstanding and the manuscript is potentially suitable for publication in *Nat. Commun.*, subject to the revisions described below.

1. Throughout the manuscript the Q β RNA is referred to as a “replicator” that “self-replicates”, but of course it is Q β protein that is the replicator. The RNA does not self-replicate, but rather is replicated by the protein that it encodes. Such incorrect language goes back to the original Spiegelman paper titled “An Extracellular Darwinian Experiment with a Self-Duplicating Nucleic Acid Molecule (*Proc. Natl. Acad. Sci. USA* 58:217–224, 1967). The nucleic acid was never “self-duplicating”, even if it sounds more dramatic to say so. In the first two sentences of the Results section, Mizuuchi et al. say it exactly right, but in the Introduction and elsewhere in the manuscript the phrasing needs to be corrected.

2. Recognizing that the fitness of an RNA species is determined both by the ability of the RNA to serve as an efficient substrate for the replicase protein and (for hosts) to encode an efficient protein, Mizuuchi et al. analyze the ability of particular proteins to replicate particular RNAs. They perform a set of “translation-uncoupled” experiments that are highly revealing and that convincingly demonstrate the interdependencies among the various replicating species over time (shown in Figure 4). Disappointingly, however, the reader never learns about the underlying biochemical properties that are responsible for these observations. The assays in these experiment involve a single 2-hour timepoint and do not separate the efficiency of translation, from the activity of the replicase protein, and from the ability of the RNA (both plus and minus strand) to serve as an efficient substrate. As a result, the manuscript is mainly about recording the details of a historical event rather than illuminating the causes of those observations. Similarly, the translation coupled experiments (shown in Figure S13) report the observed phenotypes, but do not address the underlying biochemical properties responsible for those observations.

An important feature of the evolution system employed by Mizuuchi et al. is that it enables a reductionistic analysis of competition and cooperation. The compendium of mutations shown in Figure S11 may have functional consequences for the RNA and/or protein. Even mutations within the 5′- and 3′-UTR or that are synonymous within the open reading frame might affect the level of protein production by *in vitro* translation. It would be nearly impossible to explain the functional consequences of these many mutations, but the authors should look more carefully at the key genotypes they studied (e.g., HL1-228, HL2-228, HL3-228, PL2-228, and PL3-228). They should fully uncouple translation and

replication by carrying out the following experiments: 1) using a defined concentration of clonal RNA, produced by in vitro transcription, determine the corresponding yield of protein from in vitro translation; 2) using a defined amount of purified protein, determine its ability to replicate the various RNA species. These experiments will require substantial effort, but only a small amount of effort compared to what has already been invested in this study, and the payoff is likely to justify that added effort.

3. The first paragraph of the Discussion makes the claim that this study is relevant “to understand a possible scenario toward the emergence of life” and that “pursuing complexification processes of an evolving molecular network will offer insights to understand key transitions that would have culminated in the origins of life”. These statements are true in only a very tangential way. Obviously an in vitro evolution system that employs a reconstituted E. coli translation system and an RNA and protein derived from a bacteriophage has no direct relevance to the origins of life. Presumably the authors mean to say that the type of evolving replicator networks they have observed may be analogous to what transpired during the early history of life. A more compelling argument could be made that the present study helps one understand evolutionary processes that are relevant to viral and microbial ecosystems in biology. The parallel is closer to Lenski’s microbial evolution experiments than to the RNA world. The text should be revised accordingly.

Reviewer #3 (Remarks to the Author):

This is a very interesting paper that I hope to see published. It deals with one of the big challenges of synthetic evolution: how to evolve complex catalytic networks of selfreplicating molecular agents. As pointed out by the authors, previous attempts have failed to achieve such result due to a number of constraints associated to the limitations imposed by experimental setups. Such limitations have been overcome here, and I like the simplicity and elegant design, grounded on previous work involving RNA chains that encode RNA replicases. Both the experimental results and the modelling approximations strongly support the conclusions.

As a proof of concept approach to the emergence of complex networks of cooperative replicators, this paper will be a very influential one. The literature in this area has successfully presented the theoretical basis and many precursors of what the authors now have been to deliver. I am sure this will inspire further work and I am excited to think in ways to expand the current results in other directions.

Point-by-point response

Reviewer 1

This manuscript has high potential. Very important is that is is an experimental approach to an important problem that was largely explored theoretically before. There are a few burning issues that MUST be addressed:

Response:

We are grateful for this comment. We have carefully addressed all the comments raised by the reviewer.

Comment 1-1:

1. How do you handle plus and minus strands? This is important because this could shed light on the evolution of strand asymmetry also for those species that are being translated (grossly in line with previous theoretical works). Can it be detected that translated strands are transcribed more efficiently from the untranslated ones by the replicase than the other way round?

Response 1-1:

We did not investigate the replication of the plus and minus strands separately in the original manuscript. However, it is possible to distinguish between them by modifying the method for quantitative RT-PCR. To test whether the plus and minus strands were replicated differently, we have performed translation-coupled replication experiments for each of the three host and two parasitic RNA clones at round 228 (HL1-, HL2-, HL3-, PL2-, and PL3-228) and measured the extent of each strand replication separately. In the experiments, the host RNAs were replicated with their self-encoded replicases, whereas the parasitic RNAs (PL2- and PL3-228) were replicated in the presence of a host RNA that replicates each parasitic RNA most efficiently (HL2-228 for PL2-228 and HL1-228 for PL3-228). As shown below (Supplementary Fig. S19), we found that more plus strands were synthesized than minus strands during the replication for any of the five RNAs. In other words, minus strands were more frequently used as templates than plus strands (which could be translated) by replicases. One possible explanation for this observation is the competition between replicases and the translation machinery (see Response 1-2).

Fig. S19 | Synthesis of plus and minus strands during translation-coupled RNA replication. Each RNA clone at round 228 (10 nM) was incubated at 37 °C for 5 h in the translation system. PL2- and PL3-228 were incubated in the presence of host RNA clones that replicated each RNA most efficiently (HL2-228 for PL2-228 and HL1-228 for PL3-228). The amounts of synthesized plus and minus strand RNAs at 2 and 5 h were measured by strand-specific RT-qPCR. Error bars indicate standard errors (n = 3). Measurements were taken from distinct samples.

To describe these points, we added the following paragraph in the Results section (subsection “Development of a multiple replicator network”).

“We also note that the experiments described above measured total RNA replication without separating the plus and minus strands. The efficiency of replication can vary depending on which strand is used as a template. Therefore, we measured the synthesis of plus and minus strands separately for the five RNA clones at round 228 and found that plus strands were more synthesized than minus strands for any of the RNAs (Supplementary Fig. S19).”

We also added the following sentences in the Methods section (subsection “Translation-coupled replication experiments”).

“In some cases, the synthesized amount of plus and minus strand RNAs were separately

measured by quantitative PCR using TB Green Premix Ex Taq II (Tli RNaseH Plus) (Takara) after reverse transcription with PrimeScript reverse transcriptase (Takara) and strand-specific primers (Supplementary Table 3).”

The list of primers (Supplementary Table 3) was updated accordingly.

Comment 1-2:

2. Since we are lacking a generalized replicase ribozyme you must go through a protein replicase. This is a bit unfortunate, but currently there is no better solution. BUT an extra problem is how the RNA strands interact with the in vitro translation machinery, about which we do not know much (this is also related to the first question above). Some discussion would be in order.

Response 1-2:

As the reviewer pointed out, the presence of the in vitro translation machinery would complicate RNA replication. In particular, Q β replicase and the ribosome use plus RNA strands in opposite directions, and their competition may inhibit replication. Such competition has been observed in RNA viruses (e.g., Gamarnik & Andino, *Genes Dev.*, 1998) and suggested for a host RNA (Mizuuchi et al., *ACS Synth. Biol.*, 2015). We believe that the competition between Q β replicase and the ribosome may have caused biased plus-strand synthesis during RNA replication of the clones at round 228 (see Response 1-1). To describe these points, we added the following paragraph to the Discussion section. We also note that our new experiment (Supplementary Fig. S13a) did not show detectable translational activity of the parasitic RNAs. However, there remains a possibility that there was translation of small proteins (<10 kDa) that could not be detected by our gel analysis. Future experiments, such as mass spectrometry analysis, could further examine this possibility.

“We note that in the RNA replication system, the translation process could affect RNA replication. For example, the competition between the ribosome and Q β replicase to use plus RNA strands in opposite directions may inhibit the synthesis of minus strands by replication, as previously suggested for a host RNA³⁴. Such competition may have biased the replication of RNA clones at round 228 to the plus-strand synthesis (Supplementary Fig. S19).”

Fig. S13 | Translation activity of RNA clones at round 228. **a**, Protein translation was analyzed by SDS-PAGE after incubation of each RNA clone (300 nM) at 37 °C for 2 h with a fluorescently labeled lysyl-tRNA. An example of an analyzed fluorescent gel image is displayed, sided with a trimmed white-light image of the same gel (Lane “Marker”) to visualize the pre-stained molecular weight (Mw) marker (right). The expected bands of the replicase subunit (~64 kDa) are indicated by the black arrow. Translated proteins from the parasitic RNAs were possibly undetectable with this experimental setup.

Comment 1-3:

3. RNA structures are badly missing, though knowledge about them might aid understanding considerably. This is true for the primary sequence, but with help of the Vienna package a tentative exploration of secondary structures is also feasible.

Response 1-3:

As per the reviewer’s suggestion, we used the Vienna package to predict the secondary structures of all the analyzed RNA clones. The predicted structures of the plus and minus strands are shown in Supplementary Figs. S15 and S16, respectively. Unsurprisingly, host RNA clones in the same lineages are likely to fold into similar structures, and clones in different lineages (e.g., HL1-228, HL2-228, and HL3-228) show relatively different structures, which may partly explain their specificities to different replicases. However, we also found that the clones in PL2 and PL3 (especially PL2-228 and PL3-228) showed similar structures with only minor modifications; thus, their markedly different preferences for replicases (Fig. 3d) potentially relies on sequences themselves or only slight structural changes in one or a few regions. The RNA structure data are now

described in a new paragraph of the Results section (subsection “Development of a multiple replicator network” section, where we investigated the underlying biochemical causes of the replication relationship (Fig. 3d)) as follows (see also Response 2-2). We also note that the RNA sequences of the clones are available in the source data.

“The change in template specificity possibly relied on the different properties of replicases and RNAs, such as secondary structures (Supplementary Fig. S15 and S16).”

Fig. S15 | Typical secondary structures of the RNA clones (plus strands). Centroid structures predicted by ViennaRNA⁴ are shown. Colors indicate the probability of base pairing, from purple to red (more probable). RNA sequences are available in Source Data.

Fig. S16 | Typical secondary structures of the RNA clones (minus strands). Centroid structures predicted by ViennaRNA⁴ are shown. Colors indicate the probability of base pairing, from purple to red (more probable). RNA sequences are available in Source Data.

Comment 1-4:

4. Parasites are not only variants that are translationally inert, but also those that produce dysfunctional proteins -- what about the latter in the experiment?

Response 1-4:

As pointed out by the reviewer, RNAs that produce dysfunctional proteins could also be considered parasites. One can expect that such mutants may have been abundant in the serial transfer experiments due to the high mutation rate of Q β replicase (and hence random mutations). In our previous serial transfer experiment using a similar RNA replication system (Mizuuchi & Ichihashi, *Nat. Ecol. Evol.*, 2018), we extensively

investigated host RNAs that encode dysfunctional replicases. We found that approximately 60% of the host RNA population did not produce functional replicases, likely due to random mutations. Therefore, we suspect that a substantial fraction of the population in the presented serial transfer experiments have also lost the ability to produce functional replicases and behaved similar to parasites during evolution. To discuss these points, we added the following paragraph to the Discussion section.

“In the present study, we defined parasitic RNAs as shortened sequences that deleted the replicase gene and can replicate only in the presence of a replicase-encoding RNA (host RNA). Similarly, some of the host RNAs in the population may produce dysfunctional replicases and replicate only by utilizing active replicases translated from other host RNAs. Our previous transfer experiment showed that approximately 60% of a host RNA population did not produce functional replicases due to random mutations³¹. Although we focused only on dominant mutations in this study, heterogeneity in replicase activity may be an important factor to consider for a comprehensive understanding of the evolutionary dynamics.”

Comment 1-5:

5. Interpretation of the emerging replicator networks and their stability is not clear enough. Since the protein enzyme is in the loop, I would have modelled the systems with RNA AND encoded replicase explicitly. (This raises questions of how the protein enzymes evolves in terms of structure--not addresses at all). This would immediately strand can compete by being faster transcribed, but also by a higher rate of attachment to the replixase also. This is pertinent to the "Four-minus-one" experiments also. It is obvious why removal of HL2 causes PL2 to go extinct. It is much less so why removal of parasite PL3 causes the extinction of HL2. The explanation can be that when you remove PL3 then HL1 gets a big advantage because PL3 is its sole parasite, whereas the generalist HL2 still suffers from the strong parasite PL2. The balance is tilted by the fact that in effect HL1 ourcomoetes HL2. Discuss!

Response 1-5:

We apologize for our unclear interpretation of the emerging replicator networks and their stability. In the original manuscript, we did not explicitly incorporate replicases in our model because it increased the number of uncertain parameters. However, we have modified the original model and constructed one in which RNA concentration is described as a function of both RNA and replicase concentrations in the same

compartment, and the replicases are translated from an RNA. The differential equations are as follows.

$$\frac{d[\text{RNA}_i]}{dt} = [\text{RNA}_i] \sum_{j=1}^5 k_{ij} [\text{Rep}_j] \left(1 - \frac{\sum_{i=1}^5 [\text{RNA}_i]}{C}\right)$$

and

$$\frac{d[\text{Rep}_i]}{dt} = k_i^t [\text{RNA}_i] \left(1 - \frac{\sum_{i=1}^5 [\text{Rep}_i]}{C^t}\right),$$

where $[\text{Rep}_i]$, k_i^t , and C^t are the concentration of the replicase translated from RNA_i in each compartment, the rate constant of replicase translation for RNA_i , and the carrying capacity for translation, respectively. Other parameters were not modified or added. k_i^t was set to 1 ($1 \leq i \leq 3$, host RNAs) or 0 ($4 \leq i \leq 5$, parasitic RNAs) as the original rate constant (k_{ij} , determined based on experiments) encompasses the translation activity. C^t was set to 30. Using the extended model, we simulated the continuous replication of the five RNAs (based on HL1-, HL2-, HL3-, PL2-, and PL3-228) and obtained similar concentration dynamics to those based on the original simpler model (Fig. 5b and Supplementary Fig. S24, shown below). We note that we did not explicitly model the association and dissociation of an RNA and a replicase due to the lack of experimental data for the properties of replicases. However, we believe that such modification of the model has a minor effect on the dynamics because the continuous replication of the four RNAs was reproduced by the simple model described above.

Fig. S24 | Dynamics of the RNA replicator network using the extended model. The simulation was performed and displayed as that presented in Fig. 5b.

We have added the description of these extensions to Supplementary Text 4. In the Methods section, we also added “We note that the extension of the model to explicitly incorporate replicases (Supplementary Text 4 and Fig. S24) generated similar results.”

Additionally, we agree that the mechanisms behind the results of “Four-minus-one”

experiments were not sufficiently explained. In the revised manuscript, we have added the following sentences to the third paragraph of the Discussion section. We have also provided the replication rate constants used in our simulations (Supplementary Fig. S23).

“The mechanisms behind these observations could be explained as follows. (1) Removal of HL1-228 resulted in the disappearance of PL3-228 through competition with PL2-228, which adapted to HL2-228. (2) Removal of HL2-228 caused the extinction of PL2-228 because it could not replicate in the absence of HL2-228. (3) Removal of PL2-228 led to the competitive exclusion of HL1-228 by HL2-228, which replicated slightly faster and was more resistant to PL3-228. (4) Removal of PL3-228 made HL1-228 outcompete HL2-228 because the remaining parasitic RNA (PL2-228) parasitized only HL2-228 among the host RNAs. The disappearance of HL2-228 then caused the extinction of PL2-228, as its replication relied on HL2-228. Overall, all RNAs aided the replication balance of each other, and thus, the long-term coexistence.”

Comment 1-6:

6. Coexistence of HL1 and HL2. Could it be that self-inhibition by association of the plus and negative strands at high concentration contributes to this phenomenon? Also, if you attempt to take replicases explicitly into account, you are advised to look at I.R. Epstein (1979) Competitive coexistence of self-reproducing macromolecules. *J Theor Biol* 78(2):271-98. doi: 10.1016/0022-5193(79)90269-8.

I suggest major revision.

Response 1-6:

We do not believe that self-inhibition due to the association of the plus and minus RNA strands at high concentrations essentially contributed to the coexistence of RNAs in HL1 and HL2 for the following two reasons. (1) Our simulation reproduced the replication dynamics of HL1- and HL2-228 without assuming the effect of template association (Fig. 5b). (2) Even if the plus and minus RNA strands associate with each other at high concentrations, the association should also occur between HL1 and HL2 RNAs because they are typically less than 50 b (2.5%) apart by mutations (Supplementary Fig. S3). In this case, template association does not contribute to the coexistence of HL1 and HL2 by specifically inhibiting the replication of one type of the lineages.

We also thank the reviewer for informing us about the important paper, which showed that the maximum number of co-replicable species is the same as the number of

Michaelis–Menten type enzymes using steady state analysis. However, in our study, we did not use the Michaelis–Menten equation to avoid unknown parameters and for simplicity (see also Response 1-5). Our system also does not reach a steady state. Therefore, although we found the study by Epstein interesting and worth considering for our future studies, we believe that the model is not directly applicable to our results.

Reviewer 2

Mizuuchi et al. have continued their previous study (Nat. Commun. 4:2494, 2013) of the co-evolution of Q β RNA and corresponding RNA-encoded Q β replicase protein, carried out within droplets of a water-in-oil emulsion. Adding to the previous 120 rounds of growth and dilution, they now have completed a total of 240 rounds, and have observed increasingly complex population dynamics among the set of replicating species. Over the course of evolution, there are a succession of “host” replicators that retain the activity of the replicase protein, as well as “parasitic” replicators that are truncated RNAs that do not encode a functional protein but are replicated by protein that is produced by a host. This work is reminiscent of that of Lenski and colleagues concerning the longitudinal evolution of *E. coli*, but Mizuuchi et al. employ a purely biochemical system that affords more precise understanding of the properties of the competing species over time. Most notably, their study demonstrates the spontaneous emergence of cooperative, as well as competitive, replicators. The scientific quality of the work is outstanding and the manuscript is potentially suitable for publication in Nat. Commun., subject to the revisions described below.

Response:

We thank the reviewer for the positive evaluation of our manuscript. We are particularly grateful that the reviewer found that “The scientific quality of the work is outstanding.” We have addressed all the comments raised by the reviewer.

Comment 2-1:

1. Throughout the manuscript the Q β RNA is referred to as a “replicator” that “self-replicates”, but of course it is Q β protein that is the replicator. The RNA does not self-replicate, but rather is replicated by the protein that it encodes. Such incorrect language goes back to the original Spiegelman paper titled “An Extracellular Darwinian Experiment with a Self-Duplicating Nucleic Acid Molecule (Proc. Natl. Acad. Sci. USA 58:217–224, 1967). The nucleic acid was never “self-duplicating”, even if it sounds more dramatic to say so. In the first two sentences of the Results section, Mizuuchi et al. say it exactly right, but in the Introduction and elsewhere in the manuscript the phrasing needs to be corrected.

Response 2-1:

We thank the reviewer for pointing out our incorrect usage of the word “self-replication”. According to the reviewer’s suggestion, we have corrected all concerned phrases as

follows:

(L48) from “a parasitic RNA that replicates by exploiting the self-replicating “host” RNA” to “a parasitic RNA that replicates by exploiting replicases derived from other RNAs (replicase-encoding “host” RNAs)”

(L190–206, where we described replication relationships between isolated RNA clones based on the directed graphs (Fig. 3d)) We rewrote the concerned phrases throughout the paragraph as follows:

“... At round 0, the ancestral RNA (HL0) was replicated by the self-encoded replicase (“HL0 replicase”). At round 120, two host (HL1 and HL2) and one parasitic (PL2) RNAs appeared, but their interaction was limited. Both HL1 and HL2 replicated with their respective replicases without detectable interdependency. HL2 replicase also replicated PL2 as efficiently as HL2, whereas HL1 replicase did not. At rounds 155–158, another host RNA lineage (HL3) appeared, and the replication relationship became complicated. HL1 and HL2 were mainly replicated by their respective replicases, whereas HL3 was primarily replicated by HL2 replicase. PL2 was also preferentially replicated by HL2 replicase. At round 228, another parasitic RNA lineage (PL3) appeared, and the replication relationship became even more complicated. HL1 acquired the ability to utilize HL2 replicase, as well as the self-encoded replicase. HL2 replicase could replicate not only HL2 but also all other RNAs. HL3 replication still largely relied on HL2, although the dependence was weakened compared to rounds 155-158. In parasitic RNA clones, PL2 was only replicated by HL2 replicase, and its replication by the other replicases became negligible. In contrast, PL3 was replicated by all three replicases similarly, but none of the replications was as efficient as that of PL2 by HL2 replicase, indicating that PL2 is an HL2-specific parasite, whereas PL3 is a general parasite...”

(L502) from “measured at 2 h in self-replication” to “measured at 2 h in replication without other RNAs”

(L745, Fig. 4b–e, labels of each bar plot) from “self” to “w/o RNA 2”

(SI, L67) from “self-replicating RNA” to “a replicating RNA”

(SI, L71) from “each RNA replication by a specific RNA” to “each RNA replication by the replicase of a specific RNA”

(SI, L239, legend of Fig. S12) from “RNA 1 “self” replication in the absence of RNA 2” to “RNA 1 replication in the absence of RNA 2”

Comment 2-2:

2. Recognizing that the fitness of an RNA species is determined both by the ability of the RNA to serve as an efficient substrate for the replicase protein and (for hosts) to encode an efficient protein, Mizuuchi et al. analyze the ability of particular proteins to replicate particular RNAs. They perform a set of “translation-uncoupled” experiments that are highly revealing and that convincingly demonstrate the interdependencies among the various replicating species over time (shown in Figure 4). Disappointingly, however, the reader never learns about the underlying biochemical properties that are responsible for these observations. The assays in these experiment involve a single 2-hour timepoint and do not separate the efficiency of translation, from the activity of the replicase protein, and from the ability of the RNA (both plus and minus strand) to serve as an efficient substrate. As a result, the manuscript is mainly about recording the details of a historical event rather than illuminating the causes of those observations. Similarly, the translation coupled experiments (shown in Figure S13) report the observed phenotypes, but do not address the underlying biochemical properties responsible for those observations.

An important feature of the evolution system employed by Mizuuchi et al. is that it enables a reductionistic analysis of competition and cooperation. The compendium of mutations shown in Figure S11 may have functional consequences for the RNA and/or protein. Even mutations within the 5′- and 3′-UTR or that are synonymous within the open reading frame might affect the level of protein production by in vitro translation. It would be nearly impossible to explain the functional consequences of these many mutations, but the authors should look more carefully at the key genotypes they studied (e.g., HL1-228, HL2-228, HL3-228, PL2-228, and PL3-228). They should fully uncouple translation and replication by carrying out the following experiments: 1) using a defined concentration of clonal RNA, produced by in vitro transcription, determine the corresponding yield of protein from in vitro translation; 2) using a defined amount of purified protein, determine its ability to replicate the various RNA species. These experiments will require substantial effort, but only a small amount of effort compared to what has already been invested in this study, and the payoff is likely to justify that added effort.

Response 2-2:

We thank the reviewer for this important suggestion. As the reviewer pointed out, a notable feature of our system is the possibility of characterizing each component. Following the reviewer's suggestion, we have performed experiments to fully decouple translation and replication for the five RNA clones of interest (HL1-228, HL2-228, HL3-228, PL2-228, and PL3-228). First, we performed a translation reaction and analyzed the synthesized proteins by sodium dodecyl sulfate polyacrylamide gel electrophoresis (SDS-PAGE). To detect only synthesized proteins in the presence of a high concentration of translation proteins of diverse sizes in the translation system, we performed a translation reaction with fluorescently labeled lysine and obtained a fluorescent image of the gel. As shown in Supplementary Fig. S13, we found that the translation activity of HL1-228 and HL3-228 were approximately 2-fold higher than that of HL2-228, and no protein synthesis was detectable for PL2-228 and PL3-228. The translation of the original host RNA was also investigated for comparison. We note that we could not quantify the absolute concentration of an expressed protein because of the uncertainty of the number of fluorescently labeled lysine residues in each protein molecule.

Fig. S13 | Translation activity of RNA clones at round 228. **a**, Protein translation was analyzed by SDS-PAGE after incubation of each RNA clone (300 nM) at 37 °C for 2 h with a fluorescently labeled lysyl-tRNA. An example of an analyzed fluorescent gel image is displayed, sided with a trimmed white-light image of the same gel (Lane “Marker”) to visualize the pre-stained molecular weight (Mw) marker (right). The expected bands of the replicase subunit (~64 kDa) are indicated by the black arrow. Translated proteins from the parasitic RNAs were possibly undetectable with this experimental setup. **b**, Amount of synthesized replicase subunit, normalized to that of the ancestral host RNA

(HL0-0). Error bars indicate standard errors ($n = 3$). Measurements were taken from distinct samples. ND, not detected.

Next, to fully decouple replication from translation, we purified replicases encoded in HL1-228, HL2-228, and HL3-228. Purification was performed as described in our previous study (Ichihashi et al., *J. Biol. Chem.*, 2010) through the expression of the target proteins in *Escherichia coli*, followed by ammonium sulfate precipitation and two ion-exchange chromatography (Supplementary Fig. S14a, as shown below). We then replicated each of the five RNA clones at round 228 using the purified replicases. As shown in Supplementary Fig. S14b, the results were mostly consistent with those of the translation-uncoupled replication experiment (Fig. 4e). For example, HL1-228 replicase preferentially replicated HL1-228 and PL3-228 among host and parasitic RNAs, respectively. HL2-228 replicase replicated all five RNA clones. HL3-228 replicase preferentially replicated HL3-228 and PL3-228 among host and parasitic RNAs, respectively. Taken together, these results suggest that the observed interdependencies among the five RNAs at round 228 in the translation-uncoupled experiments can mostly be explained by the different template specificities of each evolved replicase encoded by the three host RNAs.

Fig. S14 | Replication of RNA clones at round 228 by their encoded replicases. a, Purification of mutant Q β replicases composed of EF-Tu, EF-Ts, and each of the catalytic subunits encoded by HL1-, HL2-, and HL3-228. The purified replicases after cation exchange chromatography were analyzed by 10% SDS-PAGE. M, molecular weight (Mw) marker; FT, flow-through fraction; BP, samples before purification; Eluted, samples eluted at the indicated times. The expected bands of the catalytic subunit (~64 kDa), EF-Tu (~43 kDa), and EF-Ts (~30 kDa) are indicated by the black arrowheads. Two separate gels were displayed as indicated. Eluted fractions including ones indicated by the black arrows were collected as purified Q β replicases. **b,** Replication of the RNA clones (10 nM) by each of the purified Q β replicases (10 nM) at 37 °C for 2 h, measured by RT-qPCR. Error bars indicate standard errors (n = 3–4). Measurements were taken from distinct samples.

To describe these points, we added the following paragraph to the Results section (subsection “Development of a multiple replicator network”).

“To characterize the biochemical properties underlying the observed replication relationship, we further examined the five RNA clones at round 228 (HL1-, HL2-, HL3-, PL2-, and PL3-228) by fully decoupling translation and replication reactions. We first analyzed the synthesis of each encoded replicase during translation and found that HL1- and HL3-228 showed approximately twice as much replicase synthesis than HL2-228, whereas PL2- and PL3-228 did not show detectable translation activity (Supplementary Fig. S13). Next, we replicated the RNA clones using purified replicases derived from HL1-, HL2-, and HL3-228 (Supplementary Fig. S14) and found that the tendency of replication was mostly consistent with that of the translation-uncoupled replication experiment (Fig. 4e). For example, HL1- and HL3-228 replicases preferentially replicated their corresponding host RNAs (HL1- and HL3-238, respectively) and PL3-228 among parasitic RNAs, whereas HL2-228 replicase replicated all five RNA clones. These results indicate that the interdependent RNA replication at round 228 (Fig. 3d) can be mainly explained by the different template specificities of the three evolved replicases. The change in template specificity possibly relied on the different properties of replicases and RNAs, such as secondary structures (Supplementary Fig. S15 and S16).”

We note that we have added the predicted secondary structures of each RNA clone as Supplementary Fig. S15 and S16 (see also Response 1-3). In addition, we added the following paragraphs in the Methods section.

“Analysis of protein translation by sodium dodecyl sulfate polyacrylamide gel electrophoresis (SDS-PAGE)

An RNA clone (300 nM) was incubated at 37 °C for 2 h in the translation system and FluoroTect GreenLys tRNA (Promega), without UTP to preclude RNA replication. After translation, an aliquot was incubated at 95 °C for 4 min in SDS sample buffer [50 mM tris(hydroxymethyl)aminomethane hydrochloride (Tris-HCl, pH 7.4), 2% SDS, 0.86 M 2-mercaptoethanol, and 10% glycerol] and subjected to 10% SDS-PAGE. The synthesized fluorescently labeled proteins were visualized using FUSION-SL4 (Vilber-Lourmat).

RNA replication by purified Q β replicase

Q β replicase of each RNA clone (HL1-228, HL2-228, and HL3-228) was purified as described in the previous study⁵⁵. Briefly, an encoded replicase subunit was co-expressed with EF-Tu and EF-Ts in *Escherichia coli*, and then, the cell lysate was subjected to ammonium sulfate precipitation, followed by anion and cation exchange chromatography. Purified replicases were analyzed by 10% SDS-PAGE and Coomassie Brilliant Blue staining. For the replication reaction, 10 nM of an RNA clone was replicated at 37 °C for 2 h using 10 nM of each purified replicase. The reaction was performed under the same conditions for the replication reaction in translation-uncoupled replication experiments, and the fold replication of each RNA clone was determined accordingly.

Comment 2-3:

3. The first paragraph of the Discussion makes the claim that this study is relevant “to understand a possible scenario toward the emergence of life” and that “pursuing complexification processes of an evolving molecular network will offer insights to understand key transitions that would have culminated in the origins of life”. These statements are true in only a very tangential way. Obviously an in vitro evolution system that employs a reconstituted *E. coli* translation system and an RNA and protein derived from a bacteriophage has no direct relevance to the origins of life. Presumably the authors mean to say that the type of evolving replicator networks they have observed may be analogous to what transpired during the early history of life. A more compelling argument could be made that the present study helps one understand evolutionary processes that are relevant to viral and microbial ecosystems in biology. The parallel is closer to Lenski’s microbial evolution experiments than to the RNA world. The text should be revised accordingly.

Response 2-3:

We believe that the reviewer has referred to the last paragraph of the Discussion section. As the reviewer pointed out, we believed that “the type of evolving replicator networks (we) have observed may be analogous to what transpired during the early history of life.” In the original manuscript, we refrained from stating the relevance of our system to viral and microbial ecosystems in biology because our system is artificial and much simpler than viral and microbial systems. However, with the reviewer’s supportive push, we rewrote the concerned paragraph by removing the statement about the relevance to the origins of life and instead discussed the potential relevance to the evolution of biological systems (including Lenski’s microbial evolution experiments) as follows.

“Long-term experimental evolution is a powerful methodology that has provided us with fundamental insights into the principle of evolution^{41–44}. Laboratory evolution of bacterial, eukaryotic, and viral systems has also demonstrated host-parasite coevolution and diversification^{39,45–51}, similar to those observed in this study. The simplicity of our RNA replication system, compared with biological organisms, allows us to examine evolutionary events with unprecedented resolution. For example, a small set of defined components enables the detailed characterization of replication strategies for each species (e.g., Fig. 4, Supplementary Figs. S13 and S14). We can also readily obtain a large number ($> 10^5$) of the entire genome sequences of all replicating species (Supplementary Table 1) for extensive investigation of population genetics throughout evolution, which is challenging for living organisms because of the much larger genomes^{42,43}. Thus, our simple experimental setup offers a unique approach to deeply look into evolutionary phenomena.”

Reviewer 3

This is a very interesting paper that I hope to see published. It deals with one of the big challenges of synthetic evolution: how to evolve complex catalytic networks of selfreplicating molecular agents. As pointed out by the authors, previous attempts have failed to achieve such result due to a number of constraints associated to the limitations imposed by experimental setups. Such limitations have been overcome here, and I like the simplicity and elegant design, grounded on previous work involving RNA chains that encode RNA replicases. Both the experimental results and the modelling approximations strongly support the conclusions.

As a proof of concept approach to the emergence of complex networks of cooperative replicators, this paper will be a very influential one. The literature in this area has successfully presented the theoretical basis and many precursors of what the authors now have been to deliver. I am sure this will inspire further work and I am excited to think in ways to expand the current results in other directions.

Response:

We thank the reviewer for these very supportive comments, and we are happy that we can share our excitement about the present work and potential future contributions with the reviewer. We will continue the evolution experiments to address other questions associated with molecular replicators, such as how a novel function emerges, how multiple replicators become a single unit, and how to avoid error catastrophe.

REVIEWERS' COMMENTS

Reviewer #1 (Remarks to the Author):

This valuable paper has been duly revised. I appreciate the new results and discussion provided. The paper is ready for publication.

Eors Szathmary

Reviewer #2 (Remarks to the Author):

Mizuuchi et al. have submitted a revised manuscript that fully addresses all of the issues I (Reviewer 2) raised regarding their original submission. They also have done well in addressing the concerns of Reviewer 1. It is especially interesting to see the higher rate of synthesis of the plus compared to minus strand for the three host and two parasite RNAs. This already outstanding study has been substantially improved and I believe is highly suitable for publication in Nat. Commun.